# Therapeutic Opportunities of Disrupting Genome Integrity in Adult Diffuse Glioma

**DOI:** 10.3390/biomedicines10020332

**Published:** 2022-01-31

**Authors:** Diana Aguilar-Morante, Daniel Gómez-Cabello, Hazel Quek, Tianqing Liu, Petra Hamerlik, Yi Chieh Lim

**Affiliations:** 1Instituto de Biomedicina de Sevilla (IBiS), Hospital Universitario Virgen del Rocío/CSIC/Universidad de Sevilla, 41013 Sevilla, Spain; daguilar-ibis@us.es (D.A.-M.); dgcabello@us.es (D.G.-C.); 2QIMR Berghofer Medical Research Institute, Brisbane, QLD 4006, Australia; hazel.quek@qimrberghofer.edu.au; 3NICM Health Research Institute, Westmead, NSW 2145, Australia; michelle.tianqing.liu@gmail.com; 4Danish Cancer Society, 2100 København, Denmark; pkn@cancer.dk

**Keywords:** glioma, DNA damage response, DNA repair, synthetic lethality, precision medicine, targeted therapy, CNS tumors, molecular markers, pharmacotherapeutics

## Abstract

Adult diffuse glioma, particularly glioblastoma (GBM), is a devastating tumor of the central nervous system. The existential threat of this disease requires on-going treatment to counteract tumor progression. The present outcome is discouraging as most patients will succumb to this disease. The low cure rate is consistent with the failure of first-line therapy, radiation and temozolomide (TMZ). Even with their therapeutic mechanism of action to incur lethal DNA lesions, tumor growth remains undeterred. Delivering additional treatments only delays the inescapable development of therapeutic tolerance and disease recurrence. The urgency of establishing lifelong tumor control needs to be re-examined with a greater focus on eliminating resistance. Early genomic and transcriptome studies suggest each tumor subtype possesses a unique molecular network to safeguard genome integrity. Subsequent seminal work on post-therapy tumor progression sheds light on the involvement of DNA repair as the causative contributor for hypermutation and therapeutic failure. In this review, we will provide an overview of known molecular factors that influence the engagement of different DNA repair pathways, including targetable vulnerabilities, which can be exploited for clinical benefit with the use of specific inhibitors.

## 1. Introduction

Therapeutic resistance is a known phenomenon that continues to be a formidable foe in the search for a curative treatment. Based on the observation of recurring tumors, early investigation supports the idea that tumors are initiated and maintained by a small population of cancer cells that share similar biological traits to normal adult stem cells. This led to the discovery of a distinct subset of CD133+ cells that endows malignant brain tumors with radioresistant properties [1,2]. Evidence of their constitutive DNA damage response (DDR) signaling has shown that these glioma-derived neural stem-like (GNS) cells are capable of recapitulating the entire tumor population single-handedly despite being under intense therapeutic insults, whilst the remaining cancer cell types are devoid of this ability [1,3]. From a clinical perspective, the characteristics of GNS cells are frequently observed in non-responsive patients who have received the Stupp protocol, a first-line treatment regime that utilizes both radiotherapy and TMZ to deliver fatal DNA double stand breaks (DSBs) [4,5]. Such modalities are inadequate to eliminate tumor cells and it is not uncommon to find a residual cell population as a source of recurrence.

Such attributes are largely influenced by genetic fitness, where malignant cancer cells comprise or acquire specific alteration in the genome that permits the rewiring of signaling cascades to gain new modes of survival. In an attempt to unravel the relationship between driver mutations and resistance, large-scale molecular profiling by The Cancer Genome Atlas (TCGA) has taken up the role of defining the genomic landscape of malignant brain tumors [6]. Here, the seminal work shed light on three distinct subtypes; proneural, classical and mesenchymal [7]. Nearly all tumors with proneural subtype possess *PDGFRA* aberration and are associated with younger patients which have a high prevalence of *IDH1* mutation. For the classical subtype, amplification of *EGFR* and Chr.7 is correlated with the frequent loss of Chr.10 and *CDKN2A*. Conversely, the mesenchymal subtype has an angiogenic phenotype and is characterized by *NF1* mutation with extensive upregulation of necrotic and inflammatory factors. Moreover, the neural subtype was also identified during the initial analysis but has been withdrawn due to the presence of non-tumorigenic cells. Among the different subtypes, tumors with the mesenchymal profile have the worst prognosis, whereas the proneural subtype shows a more favorable outcome. From these studies, it is evident that GBM subtypes are reliant on various independent signaling networks to permit tumor growth. One interesting observation is the broad shift in tumor heterogeneity. Current estimates indicate two-thirds of all diagnosed GBM undergoes subtype switching upon treatment [7,8]. For instance, mutant *EGFR* is lost upon recurrence, leading to the establishment of other tumor subtypes. Likewise for the transition from the proneural to mesenchymal subtype, PDGFRA deficient tumors are subjected to NF1 loss.

The shift in gene alteration is crucial for post-therapeutic survival. In this discovery, approximately 60 different mutations in various assorted efforts can contribute to the gain of malignancy in primary GBM. The same tumor at recurrence can acquire >500 different mutations in excess as a result of treatment intervention [9]. Most of these genetic variations do not exhibit discernible patterns. However, a genome-wide association study on indels formation provided insight into the correlation of mutational burden and poor patient outcome. This work extends on the current knowledge of exploiting DNA lesions as a therapeutic strategy to eradicate cancer cells. Recurring tumors with significant indels are highly competent in diminishing treatment response [9]. Retrospective assessment of pair-matched de novo and post-treated malignant tumors have uncovered that the genetic evolutionary trajectories are a prelude to clonal replacement; therapies ablate vulnerable cancer cells to positively select for resistant clones to proliferate. It is unlikely repeated radiation or TMZ treatment in recurring GBM will attain further progression of a free-survival period. These combined studies suggest a high mutational burden is reflective of a scenario where recurrent tumors possess unique DNA repair systems resulting in fewer cancer cells being killed and reduced treatment efficacy. Modern radiotherapy techniques or DNA-damage inducing agents are inadequate to yield superior outcomes unless prohibition of specific DNA repair is carried out.

## 2. Implications of the DNA Damage Response Cascade

DNA lesion comes in various forms, including mismatch, chemical base modification, and single (SSB) and double strand break (DSB). Among the different lesions, SSB is the most frequently occurring and is characterized by the loss of individual nucleotides on one strand of the DNA. While SSBs are relatively harmless, they can be fatal during S-phase progression, causing replication fork collapse and subsequent strand breakages [10]. These DSBs lesions often originate from SSBs or exogenously from therapies (e.g., radiation) [11]. The effect of unresolved DNA breaks is catastrophic to genome integrity and has been implicated in numerous human disorders and cancers [12]. The ability with which tumor cells respond to DSBs determine their resistance (or sensitivity) to alkylating-based treatment. Hence, GBM is adept at engaging the DDR repertoire as a defensive mechanism to resolve any existing harm, including radiotherapy and TMZ (Figure 1) [13,14].

In the presence of DSBs, hierarchical recruitment of different DDR factors is initiated. The MRN (MRE11-RAD50-NBS1) complex first senses and tethers to the break site, bringing ATM along via protein interaction with NBS1. Tyrosine kinase C–Abl then activates Tip60 acetyltransferase, which methylates adjacent histone H3 lysine–9 (H3k9me3) to ensure an open–relaxed chromatin structure, which guarantees subsequent DDR factors access to the break site [15]. Co-localization of MRN and ATM permits the latter to phosphorylate adjacent histone H2AX molecules (γH2AX), which in turn recruits MDC1 through its BRCT domain [16]. Interaction between these two molecules is generally accepted as the initial process of DDR where cascade signaling begins. Recruitment of MDC1 creates a positive feedback loop for phosphorylated ATM accumulations, which is required to disseminate the damage signal by further activating H2AX molecules that extend kilobases along the chromatin and away from the break site [17]. Meanwhile, MDC1 continues to recruit E3 ubiquitin (Ub)–protein ligase RNF8 which cooperates with E2 conjugating enzyme UBC13 to attach Ub molecules onto histone H1 [18,19]. The Ub chains act as a non–proteolytic platform to recruit 53BP1 and RAP80-complex, which is essential for the engagement of DNA DSB repair [20,21].

Given that DNA lesions are resolved in a cell-cycle dependent manner, two downstream effector kinase proteins, namely CHK1 and CHK2, are essential for cell-cycle arrest. Their activation promptly degrades CDK complexes and stalls the cell-cycle for the timely removal of faulty lesions. For a rapid G1/S-phase arrest, the ATM–CHK2 axis initiates the ubiquitination of CDC25A, which prevents CDK2 and Cyclin–E interaction [22]. If DNA damage does occur during the later stage of G1, it is unlikely ATM–CHK2 activation can capture this lesion, thus allowing compromised cells to bypass the G1/S checkpoint entirely. S–phase is vital for the replication of DNA. Therefore, it is not surprising that multiple checkpoints are present to ensure adequate repair can take place. Here, DNA damage causes replication fork stall. The intra–S checkpoint will initiate either ATM–CHK2 activation or CHK1 phosphorylation through ATR, which in turn modulates the ubiquitination of CDC25A phosphatase to inhibit CDK2–Cyclin E/A. This process also prevents CDC45 from initializing the origin of replication. As the cell–cycle progress slows to a halt, DNA repair is activated to ensure the integrity of replication forks stall does not damage the genome. PARP presence at this stage also preferentially binds to the gaps of stalled–replication fork structure and assists in the recruitment of MRN complex and activates DDR signaling via ATM [23]. This process is a step–wise assembly and will require the recruitment of several mediators: MDC1, BRCA1 and FANCD2. Notably, there is an ongoing debate regarding the different DDR factors associated with S–phase. However, only one signaling pathway is well-described: ATM–MDC1–NBS1 phosphorylation of SMC1 which is essential for sister chromatid cohesion [24]. During early S–phase, active SMC1 prevents DNA synthesis to allow the repair of damaged DNA. As the cell-cycle progresses from S– to G2–phase, the CDK1–Cyclin B complex drives the final mitotic entry before cellular division. When dephosphorylated by CDC25A, WEE1 releases CDK1–Cyclin B complex, thereby allowing cells to enter mitosis [25]. In the event of DNA damage, ATM–CHK2 or ATM–ATR–CHK1 axis phosphorylates CDC25A, rendering CDK1–Cyclin B complex inactive to initiate arrest. On a caution note, checkpoint activation does not necessarily represent an absolute safe mechanism as compromised cells can continue to elute G2/M checkpoint.

When the appropriate cell-cycle arrest occurs, SSB and DSB repairs come into play where each process focuses on a specific class of DNA lesions (Figure 2). The primary focus of SSB repair is to prevent DSB formation prior to replication [26]. SSB repairs include base excision repair (BER), nucleotide excision repair (NER) and mismatch repair (MMR). BER mainly corrects for non-bulky lesions by removing the damaged base for subsequent nucleotide gap filling and ligation. NER on the other hand is exceptionally versatile in its ability to eliminate a wide range of structurally unrelated DNA lesions in a “cut and patch” approach. MMR has a similar process as NER except this pathway replaces single-nucleotide mismatches, which have escaped proofreading. This includes the resolution of insertion–deletion bases (indels), which often occurs when the replication complex moves across repetitive sequences (microsatellites). The complexity of such DNA damage (aka DSB) is fatal, and would require a high functioning redundancy system in the event that the participating repair pathway is compromised. Non–homologous end–joining (NHEJ) and homologous recombination (HR) are the two major repairs used for handling DSB. Micro-mediated end joining (MMEJ) and single-strand annealing (SSA) are subsidiary pathways that are also capable of resolving DSBs, but at the expense of introducing genome rearrangements. The repair of DSBs is dependent on whether DNA end resection occurs and is regulated by the cell-cycle process. When resection is blocked at G1, NHEJ is triggered by DNA-PKcs to restore genome integrity via blunt-end ligation and is essential for suppressing chromosomal translocations. Extensive end resection is stimulated during S– to G2–phase in a manner that activates MRN complex, CtIP, BRCA1 or BLM [27,28]. This allows HR, MMEJ and SSA to contest for resolving DSBs. HR is dependent on an intact homologous sister chromatid, so that RAD51 can initiate strand invasion and template copying. Conversely, SSA relies on RAD52–ssDNA complex to mediate repair in regions of the genome that have long repeat sequences [29]. Moreover, Polθ engages MMEJ to anneal and ligate the flanking micro-homologous sequences at the break site [30].

Notably, each of the four pathways leads to a different genetic outcome that can be manipulated by oncogenic factors to establish preferential repair. For instance, GBM patients that carry the *EGFRvIII* mutation can mediate radioresistance by hyperactivating DNA-PKcs to increase NHEJ activity [5]. Conversely, GBM with *MYC* mutation also drives the interaction between CDK18 and ATR to promote a more conducive cell-cycle environment, wherein enhanced HR can render DSBs ineffective [31]. Even fusion oncogene such as *BCR-ABL* has a positive stimulation on CtIP to augment SSA repair [32]. A reasonable postulation is that malignant progression is dependent on oncogenes to mount a strong proliferative index for tumor growth, which has an indirect consequence in mutational burden (replication stress) [33]. Hence, tumor cells required a greater dependency for DDR cascade to ensure the genome is largely intact during cell division [34].

## 3. Therapeutics against the DNA Damage Response

DDR acts in concert to protect the genome from detrimental harm. Under normal circumstances, the recruitment of various repair enzymes occurs in a DNA lesion-specific manner, yet alteration of the same process permits tumor cells to have a more aggressive stance. In the treatment of such cancers, chemical inhibition that targets the enzymatic activity or interaction of DNA repair proteins can be detrimental. The rationale of exploiting “pathway addiction” is to convert endogenous damage to fatal DNA lesions so that selective killing can be established. The benefit of utilizing this strategy also reduces collateral toxic accumulation of radiation and DNA-damaging chemotherapies to normal tissues.

The aspect of the tumor’s DDR is different because the majority has lost one or more associated pathways, leading to a higher dependency on the remaining pathway/s. They are susceptible to specific DDR inhibition (Table 1, top) as a result of a compromised signaling cascade that is essential for recruiting different downstream factors to counteract the presence of DNA lesions. Conversely, normal cells, which remain to have a full DDR capacity, are not vulnerable to DDR targeting due to pathway redundancy. At the apex of DDR signaling, MRN, KU70/80, RPA, SIRT6 and PARP1 are sensors that recognize various forms of DNA damage. Over the years, small molecular inhibitors have been developed for each respective DDR sensor with reasonable success. For instance, MYCN is a known oncogene driver in GBM, which promotes replication stress and is reliant on the MRN complex during S-phase cell-cycle to detect and initiate HR repair [35,36]. MIRIN inhibitor is developed to destabilize the MRN complex by targeting the 3′->5′ exonuclease activity of MRE11, which inexplicitly causes strand breakages in MYCN-driven tumor cells [36]. A similar principle could also be extended to the inhibition of SIRT6 in BRCA1/2 deficient tumor, resulting in post-replication DSBs [37].

In addition to DNA sensors, downstream cascade (transducers and effectors) also carries significant importance in amplifying damage signals and coordinating the appropriate response for transcriptional activation, checkpoint arrest and DNA repair. This series of events are first initiated by transducer proteins, DNA-PK, ATM and ATR. For DNA-PK, recruitment to the damage site promotes the inward sliding of KU70/80 to allow NHEJ core proteins to process (if required) and re-ligate the ends of the DNA. The versatility of this repair machinery in restoring different DSBs is a valuable target for cancer therapy. Early inhibitors such as Quercetin and LY294002 were promising, with the subsequent development of NU7427 and NU7441. Both inhibitors are potent at the nanomolar concentration. A surprising advancement is the dual DNA-PK/PI3K inhibitor, KU-0060648 with approximately a 500-fold increase in efficacy. Importantly, targeting DNA-PKcs is effective in MMR-deficient tumors, and is often identified in recurrent GBM with acquired TMZ resistance [43,51]. ATM and ATR, on the other hand, play a more diverse role in amplifying the damage signal and are not confined to activating a single DNA repair process. The former is present in all cell-cycle phases, while the latter is only active during DNA replication. KU-55933 was the first specific ATM inhibitor being developed. Subsequent improvement includes KU-60019, CP46722 and KU-59403, but none of these inhibitors are able to penetrate the blood-brain barrier (BBB) except for AstraZeneca’s experimental AZ32 compound, which was further optimized in becoming the first-in-class oral ATM inhibitor, AZD1390 [52]. In the context of therapeutic targeting, inhibition of ATM activity in PTEN-deficient tumors with an already diminished HR can prevent repair compensation, resulting in unresolved DSBs [53]. This is a highly effective strategy as *PTEN* deficiency is frequently associated with GBM development [54]. ATR is another promising target for cancer therapy. In the early years, developing ATR inhibitor was challenging. The first specific compound, VE-821, came to fruition in 2011 [55,56], followed by VX-970, an updated compound for trial evaluation [42]. Treatment with this inhibitor is context dependent, where synergistic efficacy is achieved through tumors with *P53* deficiency [56]. GBM patients with the loss of *P53* are likely to benefit from this treatment strategy.

As DDR signaling converges downstream to a more specific role, effector proteins are required to act as intermediaries. One notable example is the presence of DSBs in activating checkpoints arrest to facilitate DNA repair. The delay in cell-cycle progress allows the timely removal of lesions prior to replication or division. Depending on the nature of the injury and cell-cycle phase in which the lesions are encountered, extensive arrest could be initiated at G1/S, intra-S-phase or G2/M transition, all of which are governed by CHK1 or CHK2. Checkpoints failure is therapeutically effective at circumventing the engagement of DNA repair. Hence, independent CHK1 (UCN-01), CHK2 (CCT241533) or dual CHK1/2 (AZD7762) inhibitors have been extensively tested in GBM with promising outcome [44,45,57]. Targeting the DNA repair machinery, which is responsible for correcting damage to the DNA molecules, is also a viable option (Table 1, bottom). This approach is most effective when tumor cells are solely dependent on a specific repair pathway for survival, and normal cells can escape damage via pathway redundancy.

## 4. Alternate Strategy in Achieving Therapeutic Susceptibility

Since the completion of the Human Genome Project, advances in sequencing technology have enabled rapid discoveries on genetically targeted cancer therapies. Imatinib [58], Encorafenib [59] and Osimertinib [60] are among the few agents that have been successfully developed to target their corresponding pathways; BCR–ABL gene fusion, BRAF V600E mutation and EGFRvIII variant. While these inhibitors are effective in prolonging survival, not all GBM patients have targetable gain-of-function mutations.

From an alternate standpoint, leverage against non-oncogenic genes is an interesting concept that, when disrupted in conjunction with a tumor-specific mutation, can lead to a meaningful response. Termed as synthetic lethal interaction, it is first discovered in Drosophila, in which the occurrence of a single gene mutation is tolerable for survival, but the co-occurrence loss of an additional gene leads to non-viable offspring. The same genetic interaction has been extended to human studies by determining inactive mutated genes in a given tumor and targeting their respective gene partner. The best-studied trial for targeted cancer therapies that exploits this principle is the use of poly-ADP ribose polymerase (PARP) inhibitor against tumor with breast cancer gene 1 (*BRCA1*) or 2 (*BRCA2*) mutations [61]. In this sequence of events, *BRCA1/2* plays a vital role in HR repair which is essential for genome protection. Loss of PARP promotes replication fork stalling and is dependent on BRCA-dependent HR repair for recovery. BRCA-deficient tumors subjected to PARP inhibition promote selective killing but spare normal cells with functional *BRCA1/2*. Success of this outcome has spawned numerous attempts to identify new interacting partners with synthetic lethal properties (Table 2). The main obstacle, however, lies in GBM heterogeneity, which may require the disruption of two or more oncogenes to produce a lethal effect because of the different genetic background in individual tumor cells. Acquired mutation including post-therapy gene reversion, also present similar repercussion of suppressing synthetic lethality. In the former, somatic mutation of 53BP1 in BRCA-deficient tumor [62] has been shown to promote resistance against PARP inhibitor, whereas the latter undergoes further genetic alteration to regain *BRCA1/2* wild-type function [63].

Despite these setbacks, the strategy on synthetic lethality can be extended to target the two major traits of tumor cells: replication stress and metabolic rewiring. In secondary GBM, *IDH* mutation is associated with the frequent loss of *CDKN2A*, which permits the entry of damaged DNA into S-phase. While this process encourages the formation of driver mutations during replication, it also creates a therapeutic window to target nucleotide production. Without the basic building blocks for genome restoration, IDH-mutant GBM cells are susceptible to DNA damaged-induced cell death. Similar synthetic lethality strategies against replication stress include targeting AMBRA1-deficient tumors with CHK1 inhibitor [64] or POLD1-deficient tumors that receive ATR inhibitor [65]; both approaches are designed to compromise the genome with excessive strand breakages and without the means to engage DNA repair.

Metabolic alteration is another leading cancer hallmark that favors glycolysis to satisfy the biosynthetic demands for tumor growth. A prominent example is the IDH mutation in GBM that drives 2-hydroxylglutarate (2-HG) production for tumorigenesis. Elevation of this metabolic enzyme impacts chromatin formation by inhibiting KDM4B and inadvertently hypermethylates H3K9 to compact the genome. When DNA breaks occur within these heterochromatin regions, DDR factors are unable to access the damage sites and perform genome restoration, leaving tumor cells highly sensitive to alkylating-based treatments [73]. Targeting the redox balance is also an exceptional strategy. As GBM outgrow their blood supply and become hypoxic, their redox homeostasis is compromised by the production of reactive oxygen species (ROS). When left unchecked, it is deleterious to the genome. At the cellular level, oncogenes such as *RAS, RAC1, STAT3, BCL-2* and *MYC* are responsible for this elevation of ROS, and it is well-documented that GBM are dependent on glutathione to ensure excessive ROS is kept within tolerable limit. By disrupting the redox balance, tumor cells are vulnerable to either the increase in oxidative damage [74] or lack of antioxidant [75]. In a recent investigation, rocaglamide A (RocA) has been identified as a natural compound with specificity against prohibitin (PHB) that regulates ROS production. Targeting this protein overwhelms GBM with unresolved ROS-induced DNA damage [76]. Likewise, GBM with MMR deficiency can give rise to the formation of 8-oxoG via ROS [77]. Early evidence suggests this is related to the acquisition of TMZ resistance during standard-of-care [51,78,79]. Selective killing can be exploited in MMR-deficient tumors via DNA polymerase B or G inhibition [80]. As hypoxic GBM also require antioxidant such as glutathione to manage ROS, targeting this protein can disrupt the redox balance to achieve synthetic lethal interaction [81].

## 5. Mutational Signature in Predicting DNA Damage Response

Prospective therapeutic interventions of DDR are in support of a more dynamic approach, using the imprint of genes (or “signature”) to unveil vulnerable sites within biological processes, instead of employing classic radiologic and histological analysis. During the progress of tumor malignancy, exposure to exogenous and/or endogenous reactive agents allow individual cancer cells to accumulate somatic mutations. While these alterations have been previously ignored as non-oncogenic drivers, it is clear that passenger mutations and structural events represent underlying defects in the DNA repair machinery. These counteracting processes are jointly shaping genomic changes and are best exemplified by the interplay of nucleotide misincorporation between DNA polymerases and MMR pathway. When the fidelity of DNA synthesis is compromised, post-replicative MMR mediates damage removal by excision to provide subsequent opportunity of an error-free synthesis prior to cell division. In the event of a malfunction, MMR deficiency brings about the full spectrum of replication errors which can be annotated by the high prevalence of transversion, an interchange of purine to pyrimidine bases. The pattern of mutations bears clinical value as predictors in therapeutic targeting and can be broadly categorized using base substitution or indel to determine DDR dysregulation.

In base substitution, there are six different nucleotide modifications (C∙G→A∙T/G∙C/T∙A or T∙A→A∙T/C∙G/G∙C), which consist of 16 additional sequence changes for each neighboring base (A, C, G or T at the 5′ and 3′ end), thus giving rise to a theoretical 96 different trinucleotides combination in mutational patterns. The COSMIC platform houses a reference catalogue of all curated mutational signatures across different cancer types, including GBM [82]. Here, it is revealed that IDH-mutant tumors harbor a much higher frequency in mutational signature 3 (HR repair) and 15 (MMR) when compared to IDH-wildtype tumors [8]. GBM patients with IDH-mutation are indicative to benefit from alkylating therapy with specificity against S-phase cell-cycle where HR and MMR are the most active. The lack of repair during replication will exacerbate strand breakages resulting in DNA damage-induced cell death. Indels signature is another alternate analysis that is defined by the incorporation or loss of DNA fragments (<50 bp). It is found at one tenth the frequency and lacks the precise mutational coordination of base substitutions. Hence, short indels have been classified on the simple basis of deletion/insertion at the C or T base, while longer indels are defined as repeats with/without overlapping microhomology at deleted boundaries. This simplistic designation formed the premise of the 83 indel signatures, where most are associated with the alteration of proofreading polymerases, replication slippage, unwinding of double-stranded DNA and repair [45]. Importantly, when interrogating GBM with prior radiotherapy treatment, the poor survival outcome was linked to the increased burden of deleted DNA fragments. Investigation has implicated a specific enrichment for indel signature 8 (NHEJ repair) in both IDH-wildtype and IDH-mutant GBM [9]. With this newly acquired knowledge, it remains to be determined if mutational signatures can be categorically employed for therapeutic assessment.

## 6. Targeting DNA Damage Response from Preclinical Models to Clinical Trials

Although traditional chemotherapeutics are imperative for first-line treatment, there has been a growing interest in targeted approaches. Driven by omics advancement, the premise of precision oncology is to utilize therapeutics to address patient-to-patient tumor heterogeneity. Personalized therapeutics offer a compelling advantage over conventional one-size-fits-all treatment. Furthermore, challenges exist in areas of high attrition rates for drug development. These shortcomings can be mitigated by the use of genome editing (CRISPR, TALENs, ZFNs or homing endonucleases), or the RNA targeting (siRNA, ShRNA) toolkit, to understand functional vulnerabilities. Inhibitors based on these targetable genes can subsequently be developed. Combined with the use of preclinical models to evaluate drug safety profiles, the overall strategy will allow a more accurate prediction of success in clinical trials.

For instance, patient-derived xenograft (PDX) has been routinely adopted to evaluate patient-specific drug response, as it retains the genetic, transcriptional and histological features of the parental tumor. In this context, transplanting the biopsied specimen subcutaneously allows direct monitoring of tumor growth. Importantly, subcutaneous xenograft favors vasculature studies due to the availability of blood vessels while allowing different modes of drug administration; intratumoral, intraperitoneal and intravenous. The main drawback is the lack of BBB to observe therapeutic permeability. Intracranial PDX, on the other hand, is suited to examine tumor response in its native microenvironment but risks genetic drifting due to the need for cultivating biopsy tissue in vitro prior to implantation. Moreover, engraftment is slow, and lacks quantitative measures for malignant growth. Animal symptoms are the only reference for tumor progression. Despite its disadvantages, orthotopic xenograft closely mirrors cancer progression in humans. A third intraocular tumor model with the merits of both subcutaneous and intracranial approaches has also been proposed as an alternative means to evaluate therapeutic efficacy [83].

Regardless of the models used, PDXs are essential in providing accurate estimates of a compound’s pharmacokinetics and pharmacodynamics for allometric scaling in human studies. In phase-I trial, patients are usually enrolled instead of healthy subjects because of the risk-to-benefit factor. which favors the former due to the exhaustion of existing therapeutics, and drug assessment that involves higher cytotoxic compounds. This is relevant to DDR inhibitors, as they rely on RT and/or other alkylating agents to achieve synergism, which can lead to overlapping toxicity. One strategy for developing these inhibitors for phase-I study is to determine their BBB penetrance, followed by efforts to adjust dose intervals to limit cumulative side effects. Additional diagnostic work-up on radiographic imaging should be included for real-time assessment to better differentiate between tumor progression and/or observe unforeseen neurotoxicity by white matter changes. So far, only a handful of DDR inhibitors have entered early trials but none has progressed to phase-III (Table 3). The challenge is in the collection of pharmacodynamic data of tumor biopsies of end-stage patients. Pre-existing resistance mechanisms acquired in first-line treatment can conceal the activity of new therapeutics including the development of reliable biomarkers. These fundamental issues can be resolved by evaluating compounds in naïve tumor settings. As such, the introduction of a window-of-opportunity study or phase-0 trial is advantageous because it is positioned between preclinical and phase-I stages. Phase-0 trial allows the administration of sub-therapeutic micro-dosing of novel compounds in untreated patients prior to surgical resection, where biopsied specimens are subsequently obtained for pharmacodynamic analysis which lacks the interference of other therapies [84]. As phase-0 can expedite drug development and potentially saving 2–2.5 years in comparison with traditional approaches, including the benefit of phase-I integration, there is an increasing demand for such trial practices and it has been observed in the analysis of DDR inhibitors: Nedisertib (DNA-PK) and Adavosertib (WEE1) for GBM patients.

## 7. Conclusions

Great strides have been made in targeting DDR with newer and more potent inhibitors. The approval of PARP inhibitor represents the first DDR breakthrough for patients with *BRCA1/2* mutation. While radiotherapy and TMZ have proven to be the most effective treatment for GBM to date, there are also clear signs of progressive resistance, with accompanying treatment failure. The use of DDR inhibitors is becoming apparent as the next viable strategy for cancer therapy, both to eliminate DNA repair pathways that are responsible for counteracting treatment efficacy, and to exploit defects within the DDR cascade, which renders tumor cell dependency to a limited choice of repair pathway/s for survival. The latter is clinically attractive because of its advantage to promote selective killing while minimizing genotoxic build-up in normal cells. Additionally, DDR defects, as a result of genetic aberrations, bear prognostic values as biomarkers for targeted therapy. Apart from deciphering druggable targets, preclinical animal models are important translational tools for modern clinical trial designs in guiding patient selection, drug scheduling, and treatment response. There is no doubt that a multitude of opportunities still exist within the DDR network waiting to be exploited for GBM treatment (Figure 3).

## Figures and Tables

**Figure 1 biomedicines-10-00332-f001:**
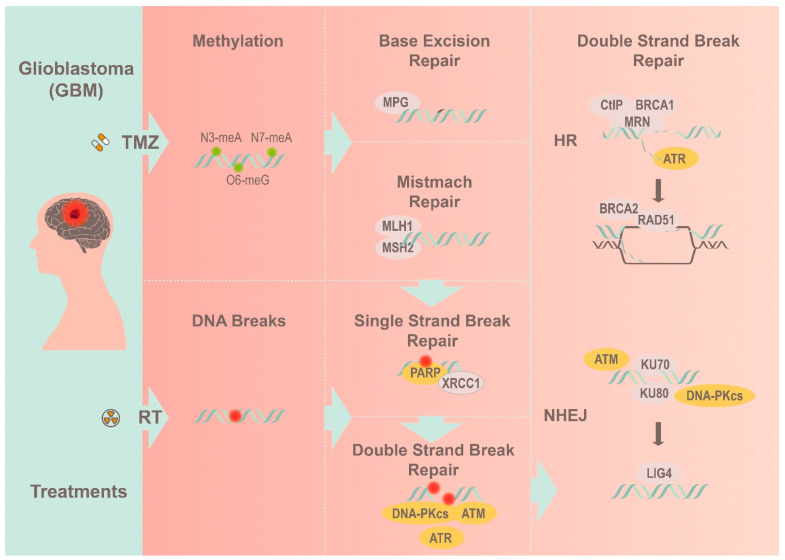
A simplified view of how GBM overcomes radiotherapy and TMZ treatment by utilizing the corresponding DNA repair pathways to resolve different DNA lesions.

**Figure 2 biomedicines-10-00332-f002:**
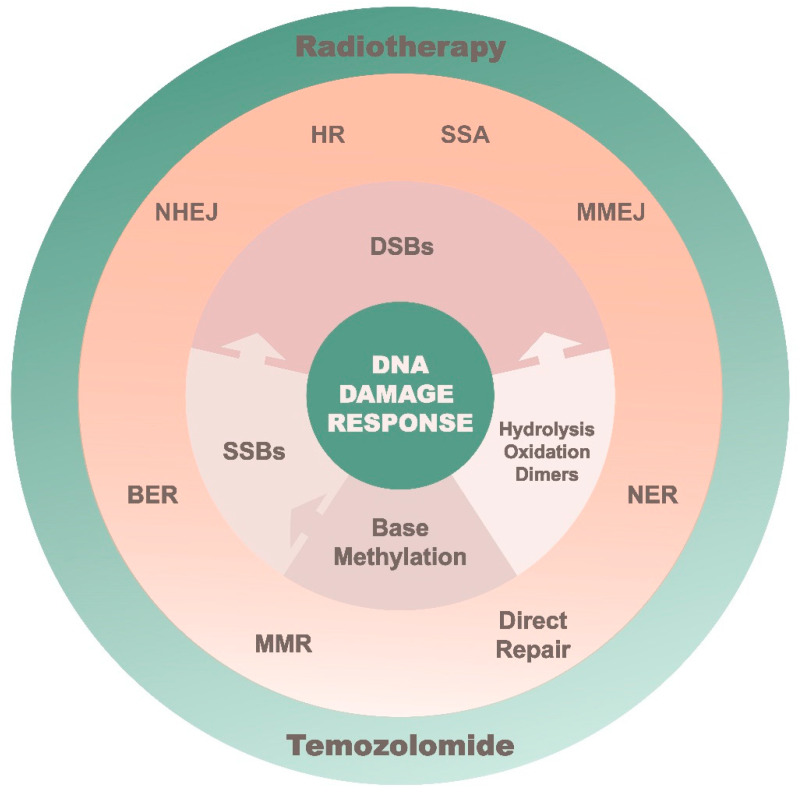
A summarized layout of the DNA repair system in combating different DNA lesions.

**Figure 3 biomedicines-10-00332-f003:**
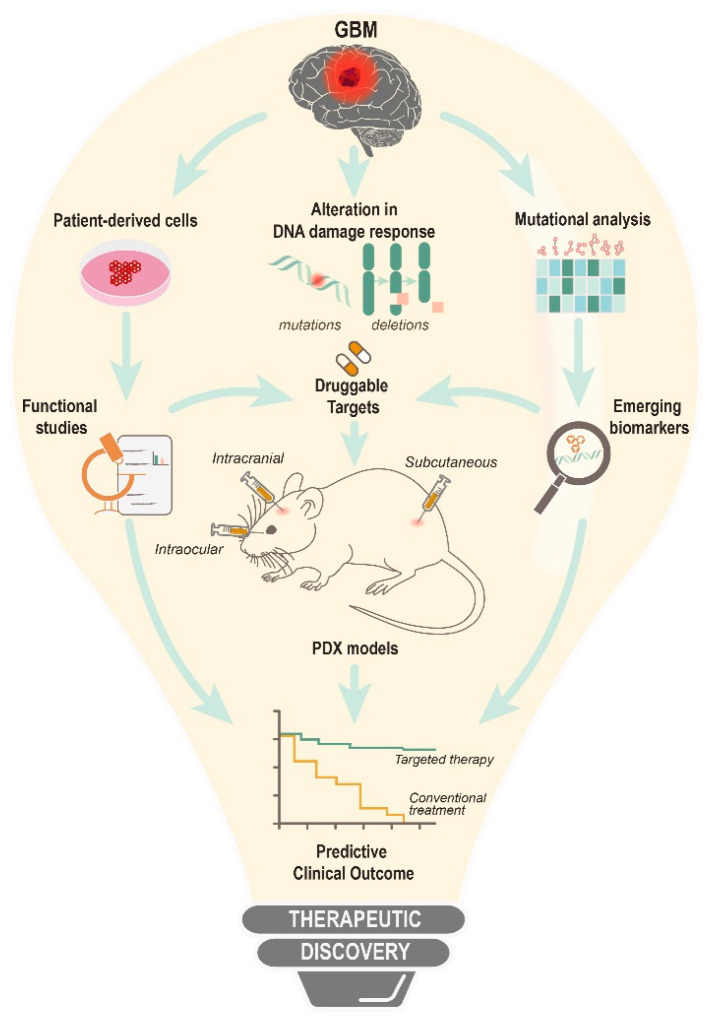
An overall view for the therapeutic discovery of DNA damage response and DNA repair inhibitors.

**Table 1 biomedicines-10-00332-t001:** Current available inhibitors of DNA damage response and DNA repair [36,38,39,40,41,42,43,44,45,46,47,48,49,50].

**DNA Damage Response Inhibitors**				
**Signaling Pathway**		**Therapeutic Agent**	**Molecular Target**	**Ref**
Sensor	MRN	Mirin	MRE11	[36]
	KU70/80	STL127705	KU70/80 and DNA-PK	[38]
	PARP1	Niraparib	PARP1 and 2	[39]
	SIRT6	SIRT6-IN-1	SIRT6,1 and 2	[40]
Transducers	ATM	KU-55933	ATM	[41]
	ATR	VX-970	ATR	[42]
	DNA-PK	KU-0060648	DNA-PK and PI3K	[43]
Effectors	CHK1	UCN-01	CHK1	[44]
	CHK2	CCT241533	CHK2	[45]
**DNA Repair Inhibitors**				
**Signaling Pathway**		**Therapeutic Agent**	**Molecular Target**	**Ref**
Double strand break repair	NHEJ	NU7026	DNA-PK	[46]
	MMEJ	ART558	Pol θ	[47]
	HR	B02	RAD51	[48]
	SSA	D-I03	RAD52	[49]
	FA	CU2	FANCL	[50]

**Table 2 biomedicines-10-00332-t002:** Interactive gene-pairs for synthetic lethality targeting [47,61,64,65,66,67,68,69,70,71,72].

Biological Process	Molecular Target	Therapeutic Inhibitor	Deficient Gene	Ref
DDR activation	ATR	VE-821	*POLD1*	[65]
DNA repair	PARP1	Niraparib	*BRCA1, BRCA2*	[61]
DNA repair	Pol θ	ART558	*BRCA1, BRCA2, FANCD2*	[47]
DNA repair	ATM	KU55933	*BRCA1*	[66]
DNA repair	DNA-PK	NU7441	*BRCA1*	[67]
DNA repair	FEN1	FEN1-IN-3	*BRCA2*	[68]
DNA sensor/repair	PARP1	Niraparib	*IDH1, IDH2*	[69]
DNA sensor/repair	PARP1	Talazoparib	*RNASEH2A, RNASEH2B*	[70]
Checkpoint arrest	CHK1	UCN-01	*AMBRA1*	[64]
Cell-cycle progression	SKP2	SKPinC1	*RB1*	[71]
Cell-cycle progression	WEE1	AZD1775	*ATRX*	[72]

**Table 3 biomedicines-10-00332-t003:** Clinical trials that use small molecule inhibitor to target the DNA damage response cascade.

Molecular Target	Inhibitor	Combination Treatment	Disease Setting	Predictive Biomarker	Clinical Phase	Clinical Trial Number
PARP1/2	Niraparib	Monotherapy	Primary GBM, recurrent GBM	*IDH* mut, *ATRX* loss	I	NCT05076513
DNA-PK	Nedisertib	RT, TMZ	GBM, Gliosarcoma	Unmethylated *MGMT*	I	NCT04555577
ATM	AZD1390	RT	GBM	-	I	NCT03423628
WEE1	Adavosertib	RT, TMZ	Primary GBM, recurrent GBM	-	I	NCT01849146
CDK4/6	Abemaciclib	Monotherapy	Recurrent GBM, Gliosarcoma	*RB*	II	NCT01227434

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
