# Peer review of "Therapeutic Opportunities of Disrupting Genome Integrity in Adult Diffuse Glioma"

_biomedicines, 2022, doi:10.3390/biomedicines10020332_

Round 1

Reviewer 1 Report

In this review article, Morante et al. have provided an overview of known factors that influence the engagement of different DNA repair pathways to address treatment strategies to improve the mortality of brain cancer patients. The authors claim that this review exemplifies extensive details of targetable enzymatic activities and complex interactions. Further, they revealed ongoing approaches to better advance the repertoire of druggable targets for clinical trial evaluation, anticipating improving GBM patient outcomes. Overall, the review is presented systematically and has tried to offer a broad detail of current advancement in the field of DDR and GBM therapy. There are certain concerns which need to be addressed to further enhance the significance of the review paper..

# Comments 1. Similar information has been published by Stagni et al. 2020, (Ferri A, Stagni V, Barilà D. Targeting the DNA Damage Response to Overcome Cancer Drug Resistance in Glioblastoma. Int J Mol Sci. 2020 Jul 11;21(14):4910. doi: 10.3390/ijms21144910. PMID: 32664581; PMCID: PMC7402284), where they have extensively discussed most of the information highlighted in the current review. Thus limiting the novelty of this paper. The author may include more discussion on the advancement of the roles of non-coding RNAs in DNA repair and their potential as biomarkers and potential therapeutic targets in the management of GBM.

# Comment 2. Discussing the advancement in the techniques including, the CRISPR-Cas9 or the RNA interference, would increase the importance, as their application in screening has identified genes involved with DSB repair mechanisms that are responsible for the TMZ-resistance.

# Comment 3. A brief discussion on the DDR-relevant genetic and epigenetic alterations identified in primary GBMs would be critical.

# Comment 4. The effects of HDAC6 inhibition on DDR signaling are missing. The expression of DDR genes, involved in repair pathways for DNA double-strand breaks, was found to be upregulated in highly malignant primary and recurrent brain tumors. A potent HDAC6 inhibitor, MPT0B291, was shown to attenuate the expression of these genes, including RAD51 and CHEK1, and was more effective in blocking homologous recombination repair in GBM cells. They uncover a regulatory network among HDAC6, Sp1, and DDR genes for drug resistance and survival of GBM cells. Furthermore, MPT0B291 may serve as a potential lead compound for GBM therapy. (Ref: Yang, WB., Wu, AC., Hsu, TI. et al. Histone deacetylase 6 acts upstream of DNA damage response activation to support the survival of glioblastoma cells. Cell Death Dis 12, 884 (2021). https://doi.org/10.1038/s41419-021-04182-w)

# Comment 6.

The recently published review paper by Nazanin Majd et al., 2020 has discussed the challenges involved in the successful development of DDR inhibitors for GBM, including the intracranial location and predicted overlapping toxicities of DDR agents with current standards of care, and proposed promising strategies to overcome these hurdles. The authors shall consider incorporating this information in their discussion.(K Majd, Timothy A Yap, Dimpy Koul, Veerakumar Balasubramaniyan, Xiaolong Li, Sabbir Khan, Katilin S Gandy, W K Alfred Yung, John F de Groot, The promise of DNA damage response inhibitors for the treatment of glioblastoma, Neuro-Oncology Advances, Volume 3, Issue 1, January-December 2021, vdab015, https://doi.org/10.1093/noajnl/vdab015)

Author Response

Dear reviewer,

Thank you for assessing our manuscript. We have now addressed your comments. Please find attached document. 

Kind regards,

Dr Yi Chieh Lim

Reviewer 2 Report

Comments and Suggestions for Authors

In this review, Diana Aguilar-Morante and colleagues provide an interesting overview of the complex heterogeneity of the involvement of DNA repair pathways in the GBM resistance therapies. Although the background described is exhaustive, many paragraphs (i.e, lines 96-142; 176-201; 212-228; 279-317 etc) are without any -text citations. Reviewing and evaluating this publication eligible for publication is challenging indeed.

The main objective to be addressed in a review, in my opinion, is to improve the knowledge of a particular topic summarizing data known from the literature. Authors should discuss the relevance of DNA repair mechanisms in tumour formation, aggression and treatment resistance, underlying data in the literature, wherein alterations in DNA repair pathways facilitate glioma progression, and how the inhibition of these repair pathways might be feasible to improve therapeutic strategies.

It is, therefore, mandatory that Authors should give an appropriate List of References to support data presented.

Moreover, the goal described in the abstract needs to be changed, (lines 20-23: with the goal of addressing treatment strategies to improve the mortality of brain cancer patients !!!!), assuming the author’s aim was to look for strategies to improve cancer cell mortality of brain cancer patients.

A table to summarize the alteration of DNA repair and their prognostic relevance in GBM, could be useful for the completeness of the review topic .

Many abbreviations were used in this manuscript, and reading the manuscript becomes difficult. Abbreviations should be added in the paper.

Minor concerns to address:

Figure 1: is not described by the text in which it is cited (lines 85-95) or in relative captation.

Table1: Authors should clarify the different meaning between "DNA Damage Response" and "DNA Repair" used in the title. The function and role of DNA repair in glioblastoma is complex and dynamic, the DNA repair factors act in response to DNA damage.

In all tables should be added the references.

Author Response

(The authors gave the same response as above.)
